# A Missense Variant in PLP2 in Holstein Cattle with X-Linked Congenital Mast Cell Tumor

**DOI:** 10.3390/ani12182329

**Published:** 2022-09-07

**Authors:** Joana G. P. Jacinto, Luisa Vera Muscatello, Irene M. Häfliger, Cinzia Benazzi, Marilena Bolcato, Arcangelo Gentile, Cord Drögemüller

**Affiliations:** 1Department of Veterinary Medical Sciences, University of Bologna, 40064 Bologna, Italy; 2Institute of Genetics, Vetsuisse Faculty, University of Bern, 3012 Bern, Switzerland

**Keywords:** bovine, cattle, mast cell tumor, congenital neoplasm, precision medicine, whole-genome sequencing

## Abstract

**Simple Summary:**

Congenital mast cell tumor is an uncommon disease in both human and veterinary medicine. In cattle, usually, such anomalies are not further investigated at the molecular genetic level, mainly because of a lack of resources and diagnostic tools and the low value and short life expectancy of the affected animals. Here we reported the clinical, pathological and genetic findings of a Holstein calf that had multiple cutaneous and visceral poorly differentiated embryonal mast cell tumors consistent with a form of congenital mast cell tumor. Whole-genome sequencing identified a single X-linked recessive protein-changing variant in the *PLP2* gene that was unique to the affected male calf and its dam and absent in a representative control cohort. Given the rarity of the identified variant, its predicted deleterious effect and the known function of proteolipid protein 2, a small transmembrane lipoprotein associated with skin cancer, we proposed a novel candidate gene associated with mast cell tumors.

**Abstract:**

Congenital tumors occur infrequently in cattle. The aim of this study was to detail the clinicopathological phenotype of a Holstein calf with a congenital mast cell tumor and to identify the genetic cause by a whole-genome sequencing (WGS) trio-approach. An 18-day-old male Holstein calf was clinically examed and revealed multifocal, alopecic, thick and wrinkled skin lesions over the entire body. At 6 months of age, the general condition of the calf was characterized by retarded growth, poor nutritional status, and ulceration of the skin lesions. Histopathological examination revealed a primary cutaneous, poorly differentiated embryonal mast cell tumor with metastases in the lymph nodes and liver. Genetic analysis revealed a private X-linked variant in the *PLP2* gene (chrX:87216480C > T; c.50C > T), which was present only in the genomes of the case (hemizygous) and his mother (heterozygous). It was absent in the sire as well as in 5365 control genomes. The identified missense variant exchanges the encoded amino acid of PLP2 at position 17 (p.Thr17Ile), which is classified as deleterious and affects a protein that plays a role in tumor growth and metastasis. Therefore, we suggested that the detected PLPL2 variant could be a plausible cause for this congenital condition in the affected calf.

## 1. Introduction

Since their first description in 1879 by Paul Ehrlich, mast cells have mainly been considered triggers of allergies [1]. Only in recent decades has the involvement of mast cells in other physiological and pathological conditions been recognized [2]. Mast cells are widely distributed in the organisms and are found primarily at the interface between the host and the external environment (skin, lungs, and gastrointestinal tract). The maturation, phenotype and function of mast cells are a direct consequence of the local microenvironment [3]. They have a marked influence on their ability to specifically recognize and respond to various stimuli by releasing several biologically active mediators [2]. These properties allow mast cells to act as first responders in harmful situations as well as to respond to changes in their environment by communicating with a variety of other cells involved in physiological and immunological responses. Therefore, mast cells have a key role in both innate and adaptive immunity. Conversely, mast cell dysfunction has been shown to play an important role in various chronic allergic, inflammatory, autoimmune and neoplastic disorders [4].

Mast cells arise from CD34 progenitor cells in the bone marrow [5]. Whereas most cells that arise from myeloid progenitor cells in the bone marrow differentiate into mature cells before leaving the bone marrow, mast cells leave the bone marrow as morphologically unidentifiable progenitor cells and differentiate into phenotypically identifiable mast cells only after infiltration of connective or mucosal tissue [6]. Mast cell proliferation is controlled by T cell-derived factors, such us IL-3 and IL-4, and fibroblast-derived factors [6]. Infiltration of mast cells is commonly observed in various tumors, and its accumulation, either at the peri-tumoral or intra-tumoral level, has been associated with both the promotion and suppression of tumor growth [7]. In the particular case of mast cell tumors, they are known to undergo a neoplastic transformation in solitary and multiple cutaneous mast cell tumors and in visceral or systemic mastocytosis [8,9]. To date, the exact molecular mechanisms involved in the accumulation of mast cells in tumors are poorly understood.

To date, few congenital cases of mast cell tumors have been reported in calves most commonly of the Holstein breed [10,11,12,13,14]. They usually are nodular masses with a random distribution that occasionally ulcerate. Some of the cases are restricted to a diffuse cutaneous form characterized by cutaneous thickness and wrinkleless without visceral involvement [12,13]. Other cases are systemic affecting the skin and visceral organs such as lung, spleen, and skeletal [11,14]. However, none of the reported cases mention a genetic cause. In dogs, cutaneous mast cell tumors are a commonly diagnosed malignancy of the skin, especially in the Boxer and related breeds, suggesting a genetic predisposition [15].

Therefore, this study aimed to characterize in detail the clinical and pathological phenotype of a Holstein calf with a congenital mast cell tumor and to verify the presence of a possible responsible genetic variant causing the disease by performing trio whole genome sequencing (WGS).

## 2. Materials and Methods

### 2.1. Clinical Investigation

An 18-days-old male purebred Holstein calf, weighting 38 kg, was admitted to the Clinic for Ruminants of the University of Bologna due to multiple skin lesions present since birth. The calf was the result of artificial insemination with a purebred Holstein sire on a Holstein dam. The parents were not related for at least four generations. The dam was a third lactation cow that had previously given birth to two healthy female calves. The affected calf was clinically examined, and a complete blood count (CBC), serum biochemical analysis, and venous blood gas analysis were obtained at recovery and repeated 5 months later (age 6 months).

At six months of age, the calf showed worsening of the general condition related to the neoplastic disorder and was euthanized because of welfare reasons. The animal was subsequently submitted for necropsy and histologic examination.

Samples of each mass were collected, formalin-fixed and paraffine-wax embedded. Three-micron thick sections were cut and routinely stained with hematoxylin and eosin (H&E) using the automatic stainer Histo-line ATS200 (Histo-line Laboratories, Milan, Italy) and with the histochemical staining toluidine blue (0.03%; MERCK and Co., Readington, NJ, USA).

Serial sections 3 microns thick were obtained and immunohistochemistry was carried out by means of antibodies against -CD3, -CD79, -synaptophysin and -chromogranin. Antibodies’ details and specifications are summarized in Table 1. Three-micron thick sections were dewaxed in diaphane and rehydrated. Endogenous peroxidase was blocked by immersion in 3% H_2_O_2_ diluted in methanol for 30 min. Antigen retrieval was performed by incubation in a pH 6.0 citrate buffer (for Synaptophysin and Chromogranin slides) and pH 8.0 EDTA buffer (for CD3 and CD79 slides) heated for 10 min in a microwave oven at 750 W. Slides were then incubated for 30 min in a blocking solution containing 10% normal goat serum in PBS. Sections were incubated overnight at 4 °C with the primary antibodies. Antibodies’ details are summarized in Table 1. Antibody binding was visualized using a commercial avidin-biotin peroxidase kit (VECTASTAIN ABC Kits, Peterborough, UK). The chromogen DAB (3, 30 diaminobenzidine) was used. Slides were counterstained with Harris’s hematoxylin. For negative controls, the primary antibody was replaced with an irrelevant, isotype-matched antibody to control for nonspecific binding of the secondary antibody. Normal healthy lymph nodes and pancreas from other bovines were used as positive controls.

### 2.2. DNA Samples

Genomic DNA was isolated from EDTA-venous blood of the affected calf and its dam sampled at the moment of admission to the clinic and from semen of its sire purchased on the market. The extraction of the DNA was realized using the Promega Maxwell RSC DNA system (Promega, Dübendorf, Switzerland).

### 2.3. Whole-Genome Sequencing and Variant Calling

WGS trio-approach using the Illumina NovaSeq6000 (Illumina Inc., San Diego, CA, USA) was performed on the genomic DNA of the affected calf, its dam and sire. The sequenced reads were aligned to the ARS-UCD1.2 reference genome [16], resulting in an average coverage of approximately 17.6× in the calf, 17.8× in its dam and 22× in its sire. Single-nucleotide variants (SNVs) and small indel variants were called. The applied software and steps to process fastq files into binary alignment map (BAM) and genomic variant call format (GVCF) files were in accordance with the processing guidelines of the 1000 Bull Genomes Project (run 7) [17], with the exception of trimming, which was performed with fastp [18]. Further processing of the genomic data was performed according to Häfliger et al. 2020 [19]. The effects of the above variants were functionally evaluated with snpeff v4.3 [20], using the NCBI Annotation Release 106 (https://www.ncbi.nlm.nih.gov/genome/annotation_euk/Bos_taurus/106/; acceded on 19 January 2022). This resulted in the final VCF file, containing individual variants and their functional annotations. To find private variants, we compared the genotypes of the case with 825 cattle genomes of different breeds sequenced as part of the ongoing Swiss Comparative Bovine Resequencing project. All of its data are available (https://www.ebi.ac.uk/ena/browser/view/PRJEB18113; accessed on 19 January 2022) in the European Nucleotide Archive (SAMEA7015112, SAMEA7690234, SAMEA7690235 are the samples accession number of the affected calf, its dam and its sire respectively). Integrative Genomics Viewer (IGV) [21] software version 2.0 (Broad institute, Cambridge, MA, USA) was used for visual evaluation of genome regions containing potential candidate genes.

### 2.4. Validation and Selection of Potential Candidate Variants

#### 2.4.1. Occurrence of Variants in a Global Control Cohort

The comprehensive variant catalog of run 9 of the 1000 Bull Genomes Project was available to investigate the allele distribution of variants within a global control cohort [17]. The whole data set includes 5116 cattle genomes including 576 from the Swiss Comparative Bovine Resequencing project, from a variety of breeds (>130 breeds indicated). Within the dataset, there are 1209 purebred Holstein cattle, allowing for the exclusion of common variants.

#### 2.4.2. Evaluation of the Molecular Consequences of Amino Acid Substitutions

PredictSNP1 [22], PolyPhen-2 [23] and SIFT [24] were used to predict the biological consequences of the detected missense variant. For cross-species sequence alignments, the following NCBI protein accessions were considered: XP_005642144.1 (*Bos taurus*), NP_002659.1 (*Homo sapiens*), XP_001106080.1 (*Macaca mulatta*), NP_062729.1 (*Mus musculus*), NP_997484.1 (*Rattus norvegicus*), NP_001154917.1 (*Danio rerio*) and NP_001037892.1 (*Xenopus tropicalis*).

### 2.5. Sequence Accessions

All references to the bovine *PLP2* gene correspond to the NCBI accessions NC_037357.1 (chromosome X, ARS-UCD1.2), NM_203363.1 (*PLP2* mRNA), and NP_976239.1 (PLP2 protein). For the protein structure of PLP2 the Uniprot database (https://www.uniprot.org/; accessed on 11 February 2022) with accession number Q6Y1E2 was used.

## 3. Results

### 3.1. Clinical Findings

On clinical examination, at the time of hospitalization, the 18-days-old calf was bright and alert but presented multiple skin lesions. Examination of the cardiovascular, respiratory, urinary, musculoskeletal, and nervous systems revealed no abnormalities. The integumentary system examination disclosed multifocal, alopecic, thick and wrinkled skin lesions over the entire body. In the head, the skin lesions were located in the left peri-ocular and nostril region, in the right facial tuberosity and malar region, and bilaterally asymmetric at the base of the ear, ranging in size from 4 to 10 cm (Figure 1a). In the neck, two skin lesions were observed in the right distal part ranging from 5 to 9 cm (Figure 1b). In the sternal region, one lesion was observed on the right side at 7 cm. In the thoracic limbs, skin lesions were perceived in the left axial middle phalange and carpus, and in the right triceps brachii region and shoulder ranging from 3 to 13 cm. In the pelvic limbs, in the left abaxial portion of the tarsal joint and in right the plantar portion of the third phalange and metatarsal bone ranging from 1 to 13 cm (Figure 2a). Skin lesions were observed in the interdigital space of all limbs (Figure 2b) as well as in the inguinal region.

At the age of six months the calf showed worsening of the general condition characterized by retarded growth, poor nutritional condition and enlargement and ulceration of the lesions described above (Figure 3a,b).

CBC, serum biochemical analysis, and venous blood gas analysis resulted in the physiological values both at 18 days and 6 months of age.

### 3.2. Pathological Findings

On external examination the skin lesions clinically reported were examined. The cutaneous nodules were spread over the entire body surface, including the head, neck, thorax, abdomen, and limbs. The neoformations had a tendency to confluence, were roundish, raised, and the larger were ulcerated. The size ranged from 2 to 25 cm, multilobular, poorly demarcated and infiltrating the underlying subcutis. On cut sections, multiple areas had a whitish appearance, were firm in consistency and with multiple necrotic foci. The draining lymph nodes were enlarged, but no other macroscopically relevant alterations were observed in the other organs.

Histologically, the dermis and subcutis were expanded by multilobular periadnexal neoplasm (Figure 4a), poorly demarcated, not encapsulated, densely cellular and with infiltrative growth. The neoplasm was composed of lobules and solid sheets of round cells, sustained by a scant amount of fibrovascular stroma. In some sections, the cutaneous neoplastic cells had a polygonal shape. The cells were 15–20 microns in size, with distinct cell borders, an intermediate nucleus-cytoplasmic ratio and a moderate amount of eosinophilic cytoplasm. The nuclei were oval, with granular chromatin and up to 3 prominent nucleoli (Figure 4b). Anisocytosis and anisokaryosis were moderate. Mitoses were 16 in 10 high power fields (2.37 mm^2^). The dermal and subcutaneous masses presented an infiltration of eosinophils admixed with the neoplastic cells. A morphological diagnosis of undifferentiated round cell tumor was reached. Aggregated large round neoplastic cells were also detected in the subcapsular sinuses of the draining lymph node and scattered in the liver parenchyma (Figure 4c). The neoplastic cells showed a multifocal cytoplasmic purple metachromatic reaction on toluidine blue staining (Figure 4d), both in the primary cutaneous neoplasm and in the lymph nodes and liver metastases. The neoplastic cells were negative for all the immunohistochemical markers (CD3, CD79, -synaptophysin and -chromogranin, Appendix A).

Overall, the immunomorphological results were suggestive of a poorly differentiated embryonal mast cell tumor. A final diagnosis was of a congenital mast cell tumor with a primary cutaneous site and metastases in the lymph nodes and liver, compatible with a congenital mast cell tumor.

### 3.3. Genetic Analysis

Both the dam and sire of the affected calf had no clinically visible skin defects. In addition, the owner did not observe similar congenital anomalies in neonates in his herd. Therefore, it was difficult to predict whether the disorder was dominant- or recessively inherited, so we hypothesized three different possible scenarios: (I) a spontaneous, completely penetrant dominant acting de novo mutation; (II) an autosomal recessive mutation that was present in the homozygous state and inherited from both (heterozygous) parents; or (III) a maternal, X-recessive inheritance in which only the dam is a heterozygous carrier.

First, we assumed a dominantly inherited de novo variant that occurred in a single parental gamete or happened during the early embryonic development of the calf. Assuming that the causative variant is uniquely present in the affected calf and completely absent in 5365 controls, a total of 249 variants, none of which were coding, occurred privately only in the affected calf and were not present in any control (including the genomes of both parents) (Table 2).

Secondly, assuming autosomal recessive inheritance as the cause of this congenital disease, filtering revealed initially 1337 private homozygous variants in the genome of the sequenced calf that were heterozygous in both parental genomes of which five were predicted to be protein-changing (Table 2). These five variants were further evaluated for their occurrence in a global cohort of 4540 genomes from a variety of breeds revealing no unique autosomal homozygous variants exclusively present in the calf and absent in all controls (Table 2).

Finally, assuming an X-linked recessively inherited variant as the cause of this congenital disease, the first round of filtering of WGS data for protein-changing variants present hemizygous in the male calf, heterozygous in the dam and homozygous wild-type in the sire identified three variants with a predicted moderate impact. These X-chromosomal variants were further analyzed for their occurrence in a global cohort of 4540 genomes from a variety of breeds revealing a single private missense variant in *proteolipid protein 2 (PLP2)* gene exclusively present hemizygous in the genome of the affected calf and heterozygous in the genome of its dam, but absent in all other controls (Table 2). The missense variant in *PLP2* exon 1 on chromosome X (ChrX:g.87216480C > T; NM_203363.1:c.50C > T) was confirmed by visual inspection using IGV software (Figure 5). It exchanges the encoded amino acid of *PLP2* at position 17 (XP_005642144.1:p.Thr17Ile), located in the proteolipid protein 2 chain (Figure 5d). This threonine-to-isoleucine substitution affects a highly conserved residue across mammals (Figure 5c), and was predicted to be deleterious (PredictSNP1 score: 61% deleterious; PolyPhen-2 score: 45%; SIFT score: 43%).

## 4. Discussion

In this study, a comprehensive clinical, pathological and genetic study of a male Holstein calf with a congenital systemic mast cell tumor was carried out. The presented phenotype was consistent with a poorly differentiated embryonal mast cell tumor affecting the skin, the lymph nodes and the liver. Indeed, the histological examination revealed poorly differentiated neoplasms, with round cell morphology and a solid multilobular pattern. The morphological differential diagnoses were mast cell tumor, lymphoma, and an undifferentiated neuroendocrine neoplasm. The histochemical staining Toluidine Blue with a positive metachromatic reaction was diagnostic of mast cell tumor, further corroborated by the negative staining to tested immunohistochemical markers, excluding a diagnosis of lymphoma and neuroendocrine neoplasm. Few cases of congenital systemic mast cell tumor have been previously reported in Holstein and Fleckvieh calves. These cases showed a lethal (stillbirth or death shortly after birth) aggressive form of congenital systemic mast cell tumor involving several internal organs [11,14]. However, in these cases, no immunohistochemical or genetic studies were performed.

The previously reported congenital systemic mast cell tumor case in a stillborn Holstein calf was characterized by multiple lesions in the oral cavity and cutaneous lesions in the neck [11]. In addition, the previously reported congenital systemic mast cell leukosis in a Fleckvieh calf that died shortly after birth was characterized by multiple raised, cutaneous, greyish-red and partially ulcerated skin lesions all over the body [14]. The clinical and gross pathological findings of the affected calf described in this study were different from these two previous reports as the animal survived until 6 months old, had no lesions on the oral and at the first clinical examination (18-days-old) the cutaneous lesions were not ulcerated.

A possible genetic origin was considered as the cause of the observed congenital neoplasm, hypothesizing either a recessive mode of inheritance or alternatively a dominant mode of inheritance, the latter due to a spontaneous mutation, using a WGS trio-approach. Analysis of the genome sequence of the affected calf, its dam and its sire allowed the identification of a single X-linked recessively inherited coding variant exclusively present in the affected calf and absent in a global control cohort. Since no protein-changing mutation was among the more than 200 private heterozygous variants, a causal de novo variant for the observed phenotype seems rather unlikely. The identified X-linked missense variant in the *PLP2* gene was predicted to be deleterious by in silico predictions. Besides the absence in the diverse global control genome cohort, within the sub-cohort of 1209 purebred Holstein cattle, neither homozygous nor heterozygous genotypes for the deleterious allele were perceived. Considering the in silico effect prediction, the rarity of this coding variant and the known function of *PLP2* gene, we propose the identified variant as a candidate causal variant for the observed disease phenotype.

The *PLP2* missense variant affects a gene that encodes the proteolipid protein 2 playing an important role in different tumor progression (growth and metastasis) because of its capacity to promote the activation of PI3K/Akt pathway [25]. Due to this association and its increased expression, *PLP2* in cancer cells can promote cell proliferation, adhesion and invasion [26,27,28,29]. However, no association between this gene and the occurrence of mast cell tumors has yet been established in any species.

A genetic relevance has been widely highlighted in canine mast cell tumors, in which *KIT* gene mutations are significantly associated with an increased incidence of recurrent disease and death and furthermore associated with an aberrant protein localization [30]. No variants affecting the *KIT* gene were detected in the present study.

In summary, it is postulated that the congenital mast cell tumor displayed by the calf reported in this study might be associated with the identified X-chromosomal missense variant in *PLP2*. We hypothesize that the p.Thr17Ile variant has been transmitted from the maternal grandmother’s line, or first occurred as a spontaneous germline mutation during early maternal development. To prove this inheritance, genotyping of the maternal ancestors would be required. Unfortunately, samples from these animals were not available.

We consider that our findings from this case study can contribute to a better knowledge and molecular characterization of congenital systemic mast cell tumors in cattle and other mammalian species.

## 5. Conclusions

Congenital disorders such as mast cell tumors in cattle are usually unreported or misdiagnosed. The study of this case provided a complete clinical, pathological and genetic workup that allowed, for the first time, the diagnosis of a congenital systemic mast cell tumor in a calf associated with a *PLP2* missense variant. Here, we present a spontaneous large animal model for similar diseases in other mammal species and add *PLP2* to the list of candidate genes for metastatic mast cell tumors. This example illustrates the utility of precision diagnostics, including genomics, for understanding uncommon diseases and the value of monitoring cattle breeding populations for deleterious genetic diseases. Furthermore, this case study illustrates the enormous potential of genetic characterization of phenotypically well-studied cases with uncommon diseases in domestic animals such as cattle to gain new insights into the function of individual genes.

## Figures and Tables

**Figure 1 animals-12-02329-f001:**
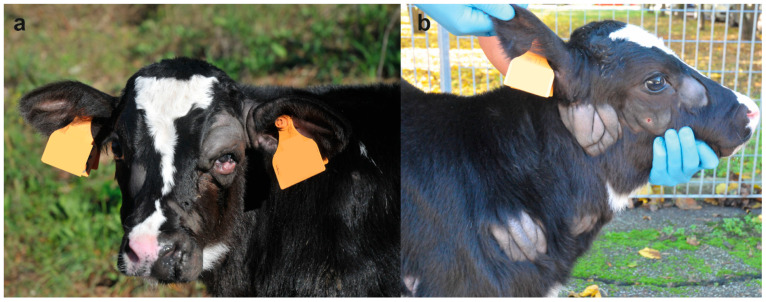
Holstein calf with congenital mast cell tumor at 18 days of age: (**a**) Note the alopecic, thick and wrinkled skin lesions in the left peri-ocular, nostril region and base of the ear ranging from 4 to 10 cm. (**b**) Note the alopecic, thick and wrinkled skin lesions in the right facial tuberosity and malar region, at the base of the ear and in the neck ranging from 4 to 10 cm.

**Figure 2 animals-12-02329-f002:**
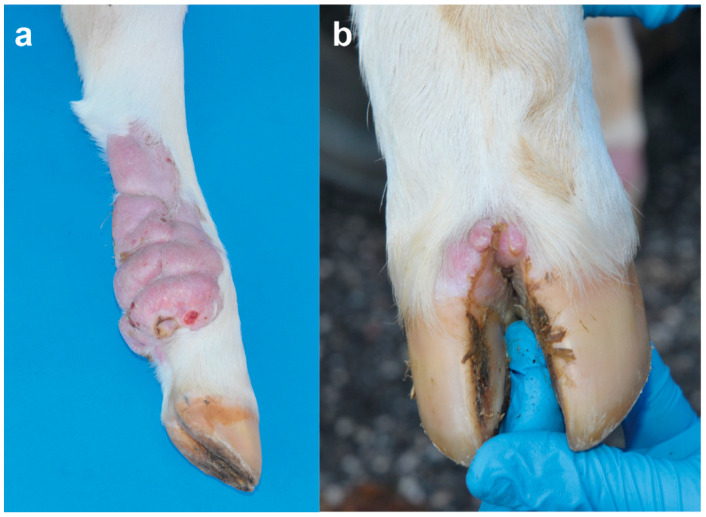
Skin lesions in the distal part of the right pelvic limb in the Holstein calf with congenital mast cell tumor at age 18 days: (**a**) Note the alopecic, thick and wrinkled skin lesions in the plantar portion of the third phalange and metatarsal bone. (**b**) Note the skin lesions in the interdigital space.

**Figure 3 animals-12-02329-f003:**
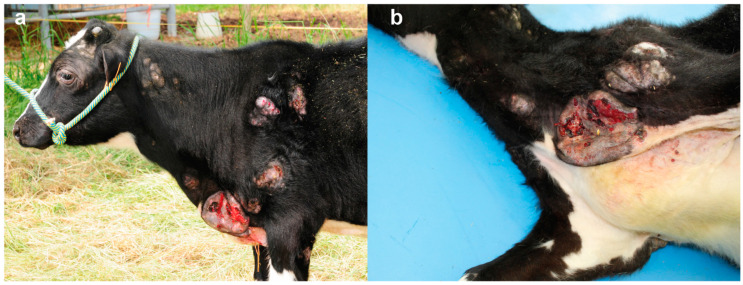
Skin lesions in the distal part of the right pelvic limb in a Holstein calf with congenital mast cell tumor at 6 months of age: (**a**) Note the ulcerated skin lesions in the neck, sternal region, and shoulder. (**b**) Magnification of the ulcerated skin lesion in the sternal region.

**Figure 4 animals-12-02329-f004:**
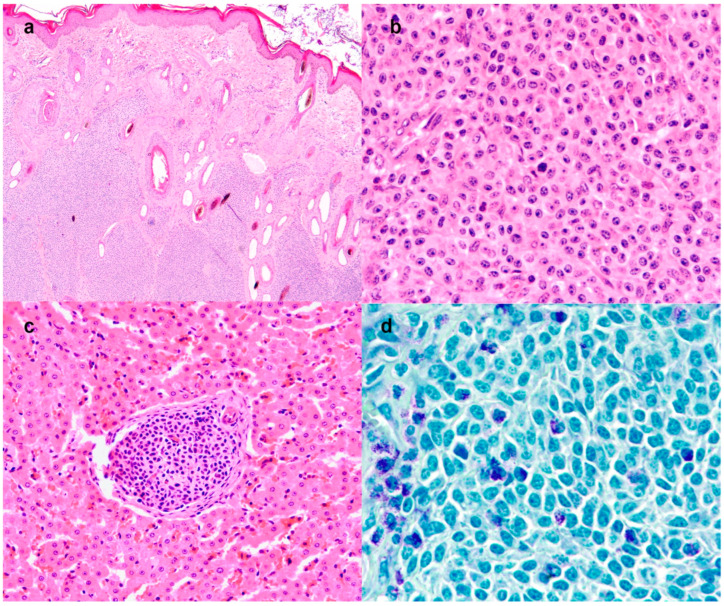
Microscopic features of the congenital mast cell tumor: (**a**) Periadnexal multilobular densely cellular poorly demarcated neoplasm, 40×, hematoxylin and eosin; **(b**) High power magnification of solid sheet of round cells with central nucleus, eosinophilic cytoplasm, and intermediate N:C ratio; scattered eosinophils are visible, 400×, hematoxylin and eosin. (**c**) Aggregate of neoplastic mast cells metastatic in the liver, 400×, hematoxylin and eosin; (**d**) Inset: metachromatic purple granules are occasionally present in the cytoplasm of neoplastic cells, toluidine blue.

**Figure 5 animals-12-02329-f005:**
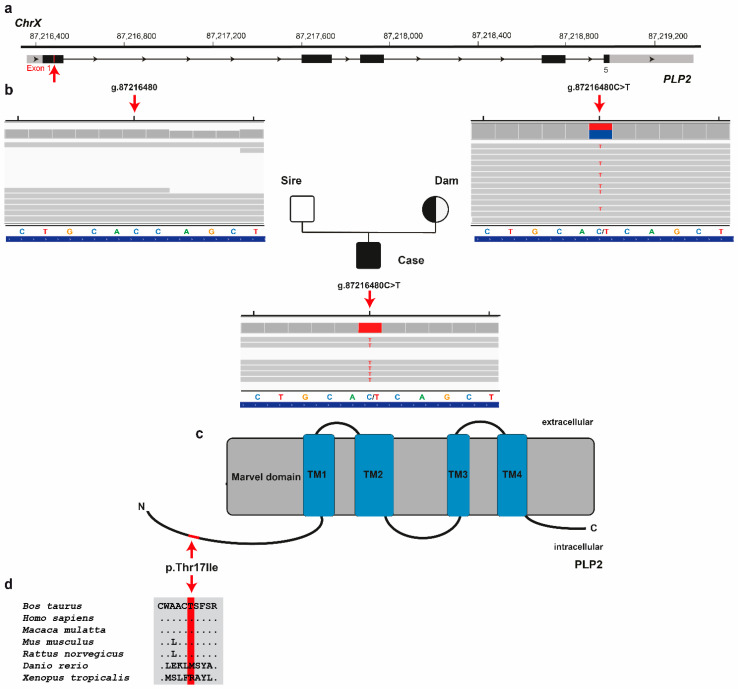
PLP2 missense variant in the male Holstein calf with a congenital mast cell tumor. (**a**) Structure of PLP2 gene showing the exon 1 variant located on chromosome X. (**b**) IGV screenshot presenting the ChrX:87216480C > T variant in the affected calf (hemizygous; shown below), its sire (homozygous wild-type; shown on top right) and its dam (heterozygous; shown on top right) revealed by whole-genome sequencing. (**c**) Schematic representation of bovine PLP2 protein and its functional domains. The red line indicates the p.Thr17Ile variant. (**d**) Cross-species sequence comparison of the proteolipid protein 2 chain with the region around the p.Thr17Ile variant shows evolutionary conservation across mammals (red indicates the affected amino acid).

**Table 1 animals-12-02329-t001:** Immunohistochemical specification and antibody details.

Antibody	Clone	Manufacturer	Dilution	Antigen Retrieval
CD3	F7.2.38	Dako, Glostrup, Denmark	1:30	EDTA buffer pH 8.0 for 10 min in microwave 750 W
CD79	HM57	Santa Cruz Biotechnology, Dallas, Texas, USA	1:400	EDTA buffer pH 8.0 for 10 min in microwave 750 W
Chromogranin	Polyclonal	Dako, Glostrup, Denmark	1:500	Citrate buffer pH 6.0 for 10 min in microwave 750 W
Synaptophysin	SY38	Dako, Glostrup, Denmark	1:50	Citrate buffer pH 6.0 for 10 min in microwave 750 W

**Table 2 animals-12-02329-t002:** Results of whole-genome sequencing variant filtering of the male Holstein calf with congenital mast cell tumor.

Filtering Step	Homozygous Variants ^1^	Heterozygous Variants
All variants in the affected calf	2,844,768	4,706,656
Private variants using 825 controls ^2^	1337	249
Private protein-changing variants using 825 controls ^2^	8 (5 autosomal, 3 X-chromosomal)	0
Remaining private protein-changing variants using a global control cohort of 4540 cattle genomes and subsequent IGV inspection	1 (X-chromosomal)	0

**^1^** Includes hemizygous variants on the X-chromosome. **^2^** Both parents were part of the 825-control cohort.

## Data Availability

The whole-genome data of the affected calf, its dam and sire are publically available at the European Nucleotide Archive (ENA) under sample accession numbers SAMEA7015112, SAMEA7690234 and SAMEA7690235, respectively.

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
