# Peer review of "A Missense Variant in PLP2 in Holstein Cattle with X-Linked Congenital Mast Cell Tumor"

_animals, 2022, doi:10.3390/ani12182329_

Round 1

Reviewer 1 Report

The manuscript is interesting and deals with a rarely published objective in veterinary medicine. Therefore I appreciate the effort made by the authors to challenge the exact characterization of congenital mast cell tumor in cattle. However, the main point is the number of animals described. As only one calf was the subject of this research, the manuscript presented no more no less like the case report. Therefore the generalizations presented at the end of the simple summary and abstract sections as well as in the conclusion section, are not justified. I strongly recommend rewriting the manuscript following the guidelines for the case reports and highlighting the nature of the manuscript both in the simple summary, abstract, and discussion sections, to avoid confusion. Moreover, the aim of the study should be revised to reflect the proper purpose of the case report. Please, see the detailed comments which intend to help you with improving the quality of your paper. As the objective is clinically interesting, I am looking to see the revised version of your manuscript.

Simple Summary

The simple summary should be a total of about 200 words maximum. As you have about 40 words left, consider expanding this section with a clear statement of the problem addressed, the aims and objectives, pertinent results, conclusions from the study, and how they will be valuable to society. 

L 15 "Here we report" - use the passive voice

L 21 "we propose a novel" - use the passive voice

Abstract

The abstract should be a total of about 200 words maximum, please, shorten it.

L 23-25 The aim should be separated from the introduction. Thus the middle part of the sentence "also known as mastocytoma in humans and many animal species" should be connected with the first sentence of the abstract, whereas, the exact aim "to detail the clinical and pathological phenotype of a Holstein calf with congenital mast cell tumor and to identify the genetic variant causing the disease by a whole-genome sequencing (WGS) trio-approach" should state separately. 

L 37 " Therefore, we suggest" - use the passive voice

Introduction

L60 As the antigen CD34 is referred to here as a characteristic of the progenitor cells in the bone marrow, consider the introduction of the characteristic antigens presented by mast cells during and after maturation, in the following part of the introduction section. Especially, that in the material and method section CD3 and CD79 are referred to.

L 71 "rare diseases" could this statement be supported by the frequency of occurrence in the population of cattle and other species including humans?

L 72 Consider expending the description of the few congenital cases of mast cell tumors reported in calves. In particular, exactly how many cases have been described and how is the incidence of the disease estimated.

L 74 "Some of the cases" how many cases from all referred? 

L 76 "Other cases" again, how many?

L 78 "In dogs, ..." what about other species? If there are no publications about them, write this, if they are - especially in cats, exotic animals, and horses - you should mention them or remove the reference to dogs. If you refer to people in the abstract and not in the introduction, consider unifying your position - removing the mention of people from the abstract or developing the topic in the introduction.

L 81-84 The aim should be rewritten, as in the case report, such kind of generalizations are not acceptable. 

L 92-94 If you do not present a complete blood count, serum biochemical analysis, and venous blood gas analysis at recovery and at age 6 months in the current paper, this sentence is unnecessary.

L 98-163 In any case, the manufacturer, city, and country of manufacture of the reagents used should be given.

L 98-100 More details on the samples proceeding protocol are required or a reference to the used standard histological protocol. 

L 98 From what exactly places samples of the masses were collected? You can use the well-described incidence of skin lesions that you presented in the results section.

L 102-103 Consider including Table S1. Immunohistochemical specification and antibody details to the main document, as it contains important data and is the only supplementary table.

L 103 The exact thickness of the microtome sections should be reported.

L 114-115 In positive controls, were the normal lymph node and pancreas collected from the affected calf or other individuals?

L 117 When and how the blood samples were collected from the affected calf and its dam?

Results

I can not find the references to figure 1 in the text.

L 189 The worsening of the general condition should be more precisely described following the same clinical protocol used at the beginning of the results section. The reader should clearly see what symptoms assessed in these two examinations indicated significant deterioration of the calf's condition.

L 192-193 I do not understand what you mean - if you did the research - present the results, if you didn't - delete this sentence.

L 208, 214, 219, 220 - throughout the whole manuscript use the same style of figures reference.

L 221-223 - provide the figure presenting the immunohistochemical features from the congenital mast cell tumors and the controls - especially the positive one.

L 224-227 This paragraph should be moved to the discussion section, as in the results section there is no place for discussing the obtained results.

L 248 Move table 1 directly after the first reference in the text. The sequence of presenting figure 5 and table 1 should be reordered.

L 285 "We performed a comprehensive" - use the passive voice

L 288 "Few cases" - as in the introduction section, please, be more precise.

L 285-292 The current clinical findings should be more precisely compared and discussed against the findings described in Smith, B.I.; Phillips, L.A. Congenital mastocytomas in a Holstein calf. Can. Vet. J. La Rev. Vet. Can. 2001, 42, 635–637 and Müller, M.; Breuer, W.; Nitzschke, K.; Drost, G.; Krisch, A.; Schmid, M.; Bogner, K.H.; Beier, D. Case report: mast cell leukosis in a neonatal calf. Dtsch. Tierarztl. Wochenschr. 2006, 113, 32–35.

L 292 Anyhow discuss your histological and immunohistochemical results in the discussion section. Since you've done your research, you shouldn't leave it without a comment

L 293 "We evaluated a possible" - use the passive voice

L 317 "we postulate that" - use the passive voice

L 328-338 The conclusion section should indicate the main conclusions or interpretations. There is no place for any introduction, speculation, or generalization when the case report is presented. Consider rewriting the conclusion section.

Author Response

Dear Editor and reviewers,

Thank you very much for your comments and suggestions that for sure will allow us to improve the quality of the article.

Reviewer 1: Comments and Suggestions for Authors

The manuscript is interesting and deals with a rarely published objective in veterinary medicine. Therefore I appreciate the effort made by the authors to challenge the exact characterization of congenital mast cell tumor in cattle. However, the main point is the number of animals described. As only one calf was the subject of this research, the manuscript presented no more no less like the case report. Therefore the generalizations presented at the end of the simple summary and abstract sections as well as in the conclusion section, are not justified. I strongly recommend rewriting the manuscript following the guidelines for the case reports and highlighting the nature of the manuscript both in the simple summary, abstract, and discussion sections, to avoid confusion. Moreover, the aim of the study should be revised to reflect the proper purpose of the case report. Please, see the detailed comments which intend to help you with improving the quality of your paper. As the objective is clinically interesting, I am looking to see the revised version of your manuscript.

Authors: Before presenting all the revision that we have done according to the reviewers’ suggestion (let say to reviewer 1 because it seems to us that reviewer 2 and 3 have accepted and considered the article acceptable without revisions!), we would like to insist on the fact that the article – although start from only one clinical case - cannot be considered a case report.

In fact, as emphasized also by reviewers 2 and 3, the importance of the article rely mainly in the demonstration that TRIO-WGS-method is an efficient approach to find out the genetic mutation of rare diseases, also if they are so rare to be occurring in only one animal (as it was our case). The clinical and morphological description of the diseased animal is the starting point for a genomic study that goes further a simple case report  with the clinical findings . Moreover, if the disease is rare, it has no sense to postpone the etiological description (described here for the first time) once another case will be reported, eventually (perhaps) many years ahead. Reviewer 3 comments: “The approach documented helps others answer the question how many cases are enough to be certain or to attempt discovery of novel mutations. In this case on family trio and existing genomic databases are adequate”.

Moreover, the limited space imposed by reformatting the article in a case report might jeopardize another important aspect of the article, that is the discussion ………………..how indicated by reviewer 3” This initial data on one gene function/outcome opens doors to study the comparative medical significance of this gene across a wide array of species”.  And might jeopardize also the description of the clinical and morphological aspects that – as indicated by reviewer -  seems to be quite adequate to the quality of the journal “Figures are excellent and informative. Histologic images are very good and useful to pathologist that may encounter this manuscript”.

Last but not least, in recent past similar comprehensive studies based on single case (which - as already mentioned - is quite often the number you can expect for rare diseases without waiting for another ten years...) where accepted by this same journal without exception:

For example:

  1. A Heterozygous Missense Variant in the COL5A2 in Holstein Cattle Resembling the Classical Ehlers–Danlos Syndrome. Animals 2020, 10(11), 2002; https://doi.org/10.3390/ani10112002;
  2. Clinicopathological and Genomic Characterization of a Simmental Calf with Generalized Bovine Juvenile Angiomatosis. Animals 2021, 11(3), 624; https://doi.org/10.3390/ani11030624
  3. A De Novo Mutation in COL1A1 in a Holstein Calf with Osteogenesis Imperfecta Type II. Animals 2021, 11(2), 561; https://doi.org/10.3390/ani11020561
  4. X-Linked Hypohidrotic Ectodermal Dysplasia in Crossbred Beef Cattle Due to a Large Deletion in EDA. Animals 2021, 11(3), 657; https://doi.org/10.3390/ani11030657
  5. A Heterozygous Missense Variant in MAP2K2 in a Stillborn Romagnola Calf with Skeletal-Cardio-Enteric Dysplasia. Animals 2021, 11(7), 1931; https://doi.org/10.3390/ani11071931

Simple Summary

Reviewer 1: The simple summary should be a total of about 200 words maximum. As you have about 40 words left, consider expanding this section with a clear statement of the problem addressed, the aims and objectives, pertinent results, conclusions from the study, and how they will be valuable to society. 

Authors: Thank you for your comments but we consider that the abstract complete and clear. In the abstract we address the aims of the study, the main results and the importance of the study for the scientific community.

Reviewer 1: L 15 "Here we report" - use the passive voice

Authors: corrected accordingly.

Reviewer 1: L 21 "we propose a novel" - use the passive voice

Authors: corrected accordingly

Abstract

Reviewer 1: The abstract should be a total of about 200 words maximum, please, shorten it.

Authors: The abstract have been shortened.

Reviewer 1: L 23-25 The aim should be separated from the introduction. Thus the middle part of the sentence "also known as mastocytoma in humans and many animal species" should be connected with the first sentence of the abstract, whereas, the exact aim "to detail the clinical and pathological phenotype of a Holstein calf with congenital mast cell tumor and to identify the genetic variant causing the disease by a whole-genome sequencing (WGS) trio-approach" should state separately. 

Authors: We corrected the second sentence of the abstract as following “The aim of this study was to detail the clinical and pathological phenotype of a Holstein calf with congenital mast cell tumor and to identify the genetic variant causing the disease by a whole-genome sequencing (WGS) trio-approach.”

Reviewer 1: L 37 " Therefore, we suggest" - use the passive voice

Authors: corrected accordingly.

Introduction

Reviewer 1: L60 As the antigen CD34 is referred to here as a characteristic of the progenitor cells in the bone marrow, consider the introduction of the characteristic antigens presented by mast cells during and after maturation, in the following part of the introduction section. Especially, that in the material and method section CD3 and CD79 are referred to.

Authors: thank you for your suggestion, we added the possible interaction between lymphocytes and mast cells. It is not the purpose of the study to go into the specifics of mast cell classification, which is still the subject of debate regarding markers in domestic species.

Reviewer 1: L 71 "rare diseases" could this statement be supported by the frequency of occurrence in the population of cattle and other species including humans?

Authors: Actually, in order to avoid misunderstanding in respect to this sentence we have decided to delete it. In fact, no universal definition in terminology and prevalence thresholds of rare disease has been definitively accepted yet, being the most widely used for human medicine the average prevalence threshold between 40 and 50 cases/100,000.

If this is true for the human medicine, even less precise might be the definition for animal diseases, especially in cattle, where many factors prevent the reporting of congenital diseases.

Therefore, we have replaced the term rare with uncommon in the entire text, preventing to state whether mastocytoma in cattle can be awarded the “official” terminology of rare disease.

Reviewer 1: L 72 Consider expending the description of the few congenital cases of mast cell tumors reported in calves. In particular, exactly how many cases have been described and how is the incidence of the disease estimated.

Authors:  See above. We have listed all the articles of our knowledge related to mastocytoma in calves.

Reviewer 1: L 74 "Some of the cases" how many cases from all referred? 

Authors: The related cases are indicated in the references in brackets at the end of the sentence.

Reviewer 1: L 76 "Other cases" again, how many?

Authors: The related cases are indicated in the references in brackets at the end of the sentence.

Reviewer 1: L 78 "In dogs, ..." what about other species? If there are no publications about them, write this, if they are - especially in cats, exotic animals, and horses - you should mention them or remove the reference to dogs. If you refer to people in the abstract and not in the introduction, consider unifying your position - removing the mention of people from the abstract or developing the topic in the introduction.

Authors: We referred dogs because is the specie where several studies have been carried out regarding a complete clinical, pathological and genetic characterization of mast cell tumors. Therefore, in our opinion it does not makes sense to refer all the other species where a possible genetic etiology remains uncovered.

Reviewer 1: L 81-84 The aim should be rewritten, as in the case report, such kind of generalizations are not acceptable. 

Authors: We do not agree with your statement. Our aim is very clear as written in the manuscript: “this study aimed to characterize in detail the clinical and pathological phenotype of a Holstein calf with a congenital mast cell tumor and to verify the presence of a possible responsible genetic variant causing the disease by performing trio whole genome sequencing (WGS).”

Reviewer 1: L 92-94 If you do not present a complete blood count, serum biochemical analysis, and venous blood gas analysis at recovery and at age 6 months in the current paper, this sentence is unnecessary.

Authors: we disagree, because the complete blood count, serum biochemical analysis, and venous blood gas analysis at recovery and at age 6 months was performed and therefore is of importance to report the results. We are convinced that it should be mentioned whether the clinical investigation  included ancillary hemato-biochemical check up or not. We would keep this sentence. In the result we changed the sentences as following: “CBC, serum biochemical analysis, and venous blood gas analysis resulted within the physiological values both at 18 days and 6 months of age.”

Reviewer 1: L 98-163 In any case, the manufacturer, city, and country of manufacture of the reagents used should be given.

Authors: The information are provided in Table 1.

Reviewer 1: L 98-100 More details on the samples proceeding protocol are required or a reference to the used standard histological protocol. 

Authors: The histological slides are stained using the automatic stainer. The additional details are added in the material and methods.

Reviewer 1: L 98 From what exactly places samples of the masses were collected? You can use the well-described incidence of skin lesions that you presented in the results section.

Authors: A sample of each mass have been collected; we specify it in the material and methods.

Reviewer 1: L 102-103 Consider including Table S1. Immunohistochemical specification and antibody details to the main document, as it contains important data and is the only supplementary table.

Authors: Done.

Reviewer 1: L 103 The exact thickness of the microtome sections should be reported.

Authors: We reported the thickness (3-micron) in the text.

Reviewer 1: L 114-115 In positive controls, were the normal lymph node and pancreas collected from the affected calf or other individuals?

Authors: Positive controls come from other individuals. We specify it in the text and as follows: “Normal healthy lymph node and pancreas from other bovines were used as positive controls.”

Reviewer 1: L 117 When and how the blood samples were collected from the affected calf and its dam?

Authors: We are not convinced that it is so important for a genomic analysis to know when and how the blood samples were collected. However, we changed the sentence as following: “Genomic DNA was isolated from EDTA-venous blood of the affected calf and its dam sampled at the moment of the admission to the clinic and from semen of its sire purchased on the market. The extraction of the DNA was realized using Promega Maxwell RSC DNA system (Promega, Dübendorf, Switzerland).”

Results

Reviewer 1: I can not find the references to figure 1 in the text.

Authors: we added the references of figure 1a and figure 1b in the text.

Reviewer 1: L 189 The worsening of the general condition should be more precisely described following the same clinical protocol used at the beginning of the results section. The reader should clearly see what symptoms assessed in these two examinations indicated significant deterioration of the calf's condition.

Authors: The worsening of the general condition was characterized by a retarded growth, poor nutritional condition and enlargement and ulceration of the lesions described.

Reviewer 1: L 192-193 I do not understand what you mean - if you did the research - present the results, if you didn't - delete this sentence.

Authors: We have changed the phrasing as following: “CBC, serum biochemical analysis, and venous blood gas analysis resulted within the physiological values both at 18 days and 6 months of age”. In our opinion it has no sense to detail the figure of the single parameters. The meaning is that the hemato-biochemical checkup did not show alterations in routine parameters.  

Reviewer 1: L 208, 214, 219, 220 - throughout the whole manuscript use the same style of figures reference.

Authors: Done.

Reviewer 1: L 221-223 - provide the figure presenting the immunohistochemical features from the congenital mast cell tumors and the controls - especially the positive one.

Authors: We added a supplementary figure showing the negative immunohistochemical results on bovine mast cell tumor, together with the positive controls.

Reviewer 1: L 224-227 This paragraph should be moved to the discussion section, as in the results section there is no place for discussing the obtained results.

Authors: We moved the paragraph to the discussion section as suggested.

Reviewer 1: L 248 Move table 1 directly after the first reference in the text. The sequence of presenting figure 5 and table 1 should be reordered.

Authors: Done.

Reviewer 1: L 285 "We performed a comprehensive" - use the passive voice

Authors: Done.

Reviewer 1: L 288 "Few cases" - as in the introduction section, please, be more precise.

Authors: As in the introduction, the related cases are indicated in the references in brackets at the end of the sentence.

Reviewer 1: L 285-292 The current clinical findings should be more precisely compared and discussed against the findings described in Smith, B.I.; Phillips, L.A. Congenital mastocytomas in a Holstein calf. Can. Vet. J. La Rev. Vet. Can. 2001, 42, 635–637 and Müller, M.; Breuer, W.; Nitzschke, K.; Drost, G.; Krisch, A.; Schmid, M.; Bogner, K.H.; Beier, D. Case report: mast cell leukosis in a neonatal calf. Dtsch. Tierarztl. Wochenschr. 2006, 113, 32–35.

Authors: We added the following paragraph to the discussion:

“The previously reported congenital systemic mast cell tumor case in a stillborn Holstein calf was characterized by multiple lesions in the oral cavity and cutaneous lesions in the neck [11]. In addition, the previously reported congenital systemic mast cell leukosis in a Fleckvieh calf that died shortly after birth was characterized by multiple raised, cutaneous, greyisch-red and partially ulcerated skin lesions all over the body [14]. The clinical and gross pathological findings of the affected calf described in this study were different from these two previous reports as the animal survived until 6-months-old, had no lesions on the oral and at the first clinical examination (18-days-old) the cutaneous lesions were not ulcerated.”

Reviewer 1: L 292 Anyhow discuss your histological and immunohistochemical results in the discussion section. Since you've done your research, you shouldn't leave it without a comment

Authors: The following sentence have been added in the discussion:

“Indeed, the histological examination revealed a poorly differentiated neoplasms, with a round cell morphology and a solid multilobular pattern. The morphological differential diagnoses were mast cell tumor, lymphoma, and an undifferentiated neuroendocrine neoplasm. The histochemical staining Toluidine Blue with a positive metachromatic reaction was diagnostic of mast cell tumor, further corroborated by the negative staining to tested immunohistochemical markers, excluding a diagnosis of lymphoma and neuroendocrine neoplasm.”

Reviewer 1: L 293 "We evaluated a possible" - use the passive voice

Authors: Done.

Reviewer 1: L 317 "we postulate that" - use the passive voice

Authors: Done

Reviewer 1: L 328-338 The conclusion section should indicate the main conclusions or interpretations. There is no place for any introduction, speculation, or generalization when the case report is presented. Consider rewriting the conclusion section.

Authors: we do not agree with your statement. In the conclusion we indicate our main conclusions and interpretation rather them speculation or generalization.

Reviewer 2 Report

Dear Aauthors, please make a control reading because for correction of the marked word!

1. What is the main question addressed by the research? The genetic background of the mast cell tumor in cattle.

2. Do you consider the topic original or relevant in the field, and if so, why? If the veterinarians see a case like this, they can recommend the owner not to use the cow for breeding.

3. What does it add to the subject area compared with other published material? This manuscript describes the variant of PLP2 gene in the calf and in the mother cow the first time which is involved with congenital mast cell tumor.

4. What specific improvements could the authors consider regarding the methodology? The genetic analyses.

5. Are the conclusions consistent with the evidence and arguments presented, and do they address the main question posed? This is the main sentence: "This example illustrates the utility 333 of precision diagnostics, including genomics, for understanding rare diseases and the 334 value of monitoring cattle breeding populations for deleterious genetic diseases."

6. Are the references appropriate? Yes.

Author Response

Dear Editor and reviewers,

Thank you very much for your comments and suggestions that for sure will allow us to improve the quality of the article.

Reviewer 3 Report

The manuscript nicely describes the disorder and highlights opportunities for specialized diagnostic investigations of rare events that may be inherited using the Trio-WGS approach. Well illustrated. The description of the approach and documented utility is as important to the readership as it the information regarding the rare disease. This initial data on one gene function/outcome opens doors to study the comparative medical significance of this gene across a wide array of species.  

1. What is the main question addressed by the research?

The authors demonstrate a mutation in PLPL2 is associated and likely causal for congenital mast cell tumor in cattle. This is a novel finding.

2. Do you consider the topic original or relevant in the field, and if so, why?

The authors demonstrate the trio-WGS approach is effective to identify mutations in rare diseases with a single case. This is important to demonstrate. While Mast cell tumor in cattle is not a significant disease issue to the cattle industry these findings advance comparative understanding of gene function that is likely to benefit biomedical research more broadly.

3. What does it add to the subject area compared with other published material?

The identification of the cause of congenital mast cell tumor in cattle is novel. The approach documented helps others answer the question how many cases are enough to be certain or to attempt discovery of novel mutations. In this case on family trio and existing genomic databases are adequate.

4. What specific improvements could the authors consider regarding the methodology?

Methods are well described in great detail. The project tis repeatable based on the methods and citations.

5. Are the conclusions consistent with the evidence and arguments presented, and do they address the main question posed?

Conclusions are based on the data presented. The three hypothesis regarding potential genetic causes are described and excluded or included in the discussion tied to the evidence from their sequencing and sequence databases.

6. Are the references appropriate?

Yes

7. Please include any additional comments on the tables and figures.

Figures are excellent and informative. Histologic images are very good and useful to pathologist that may encounter this manuscript.

Author Response

(The authors gave the same response as above.)
